# The Regulatory Role of the Central and Peripheral Serotonin Network on Feeding Signals in Metabolic Diseases

**DOI:** 10.3390/ijms23031600

**Published:** 2022-01-29

**Authors:** Katsunori Nonogaki

**Affiliations:** Laboratory of Diabetes and Nutrition, New Industry Creation Hatchery Center, Tohoku University, Sendai 980-8579, Japan; katsu@tohoku.ac.jp; Tel.: +81-22-795-5675

**Keywords:** serotonin, feeding signals, energy metabolism, 5-HT2CRs, FGF21, obesity, type 2 diabetes, NAFLD, Tph1

## Abstract

Central and peripheral serotonin (5-hydroxytryptamine, 5-HT) regulate feeding signals for energy metabolism. Disruption of central 5-HT signaling via 5-HT2C receptors (5-HT2CRs) induces leptin-independent hyperphagia in mice, leading to late-onset obesity, insulin resistance, and impaired glucose tolerance. 5-HT2CR mutant mice are more responsive than wild-type mice to a high-fat diet, exhibiting earlier-onset obesity and type 2 diabetes. High-fat and high-carbohydrate diets increase plasma 5-HT and fibroblast growth factor-21 (FGF21) levels. Plasma 5-HT and FGF21 levels are increased in rodents and humans with obesity, type 2 diabetes, and non-alcohol fatty liver diseases (NAFLD). The increases in plasma FGF21 and hepatic FGF21 expression precede hyperinsulinemia, insulin resistance, hyperglycemia, and weight gain in mice fed a high-fat diet. Nutritional, pharmacologic, or genetic inhibition of peripheral 5-HT synthesis via tryptophan hydroxylase 1 (Tph1) decreases hepatic FGF21 expression and plasma FGF21 levels in mice. Thus, perturbing central 5-HT signaling via 5-HT2CRs alters feeding behavior. Increased energy intake via a high-fat diet and/or high-carbohydrate diet can upregulate gut-derived 5-HT synthesis via Tph1. Peripheral 5-HT upregulates hepatic FGF21 expression and plasma FGF21 levels, leading to metabolic diseases such as obesity, insulin resistance, type 2 diabetes, and NAFLD. The 5-HT network in the brain–gut–liver axis regulates feeding signals and may be involved in the development and/or prevention of metabolic diseases.

## 1. Introduction

Serotonin (5-hydroxytryptamine, 5-HT) is a monoamine derived from tryptophan. Peripheral 5-HT is synthesized by tryptophan hydroxylase 1 (Tph1), which is mainly expressed in the enterochromaffin (EC) cells of the gut, whereas central 5-HT is synthesized by Tph2, which is predominantly expressed in the raphe nuclei of the brainstem [1]. Elucidation of the mechanisms underlying the effects of 5-HT on appetite and energy homeostasis has been facilitated by the discovery of at least 14 functionally diverse 5-HT receptor (R) subtypes [2]. Among the 5-HTR subtypes expressed in the hypothalamic regions that are implicated in regulating appetite and energy metabolism, including 5-HT1AR, 5-HT1BR, 5-HT2AR, 5-HT6R and 5-HT7R, gene knockout studies have particularly focused on 5-HT2CRs. Genetic ablation of 5-HT1AR, 5-HT2AR, 5-HT6R and 5-HT7R has no effects on food intake and body weight in mice fed a chow diet [2].

5-HT2Rs are the Gq/G11-protein coupled receptors, which activate phospholipase C, leading to the activation of the inositol triphosphate (IP3) and discylglycerol (DAG) downstream pathways (Table 1). 5-HT1Rs are the Gi/Go-proein coupled receptor, which inhibits adenylate cyclase (AC), leading to suppression of the cyclic adenosine monophospahte (cAMP) downstream pathways, whereas 5-HT4R, 5-HT6R, and 5-HT7R are the Gs-protein coupled receptor activates AC, leading to activation of cAMP downstream pathway (Table 1). 5-HT2CR transcripts are widely expressed throughout the central nervous system (CNS) and neuroaxis [2], regions thought to be highly involved in regulating appetite and energy homeostasis [2]. Although there are little reports on the distribution and functions of 5-HT2CRs in peripheral organs, 5-HT2CRs are expressed in white adipose tissue and their expression level is inversely related to that of plasma leptin, an adipocyte-derived hormone [2,3].

Pharmacologic compounds such as fenfluramine and sibutramine, which increase 5-HT release and/or inhibit 5-HT reuptake, and exhibit anti-obesity effects that appear to be mediated by 5-HT2CRs [4,5]; these drugs were withdrawn from the international market, however, due to their cardiac side effects via the activation of 5-HT2BRs. Another drug, lorcaserin, which is a selective 5-HT2CR agonist, induces clinically meaningful weight loss and was approved by the Food and Drug Administration (FDA) as a new anti-obesity agent in 2012 [6]. Due to its selectivity, lorcaserin is unlikely to increase the risk of cardiac valvulopathy associated with 5-HT2BR activation. The results of the CAMELLIA-TIMI 61 trial revealed no significant differences in major cardiovascular events between the lorcaserin-treated and placebo groups [7]. In addition, the results showed that lorcaserin reduced the risk for incident diabetes and microvascular complications in obese and overweight patients [8], and reduced the rate of new-onset or progressive renal impairment in comparison with placebo [9]. Moreover, the occurrence of cancer did not differ significantly between obese subjects treated with lorcaserin or placebo [7].

Nevertheless, the FDA requested the voluntarily withdrawal of lorcaserin from the US market in 2020, because a safety clinical trial demonstrated a possible increased occurrence of cancer in obese subjects treated with lorcaserin. The cause of the cancer is uncertain, and thus it cannot be concluded that lorcaserin contributes to the cancer risk. The FDA is now further evaluating the clinical trial results and will communicate their final conclusions and recommendations when the review has been completed. Anti-obesity drugs that act on central 5-HT systems, therefore, remain to be developed. Here, we review the original studies of the involvement of 5-HT2CRs in the regulation of feeding and energy metabolism, and provide a new concept of the 5-HT network in the brain–gut–liver axis contributing to the regulation of feeding signals.

## 2. Involvement of 5-HT2CRs in the Regulation of Feeding and Energy Homeostasis

### 2.1. 5-HT2CRs and Energy Metabolism

5-HT2CRs are not reportedly expressed in peripheral organs where cancer occurs, and 5-HT2CRs regulate feeding behavior in mice [10,11]. Mice with a genetic mutation of 5-HT2CRs display leptin-independent hyperphagia, which leads to late-onset obesity, insulin resistance, and impaired glucose tolerance [11]. Interestingly, hyperphagia precedes hyperinsulinemia and weight gain in the 5-HT2CR mutants [11]. These findings suggest that 5-HT2CRs do not directly regulate the central outflow of the vagal nerve to the pancreatic β-cells, and that central control of the autonomic nervous system and feeding can be dissociated [12,13]. In addition, 5-HT2CR mutants are more responsive than wild-type mice to high-fat and high-sucrose diets, which lead to earlier-onset obesity and type 2 diabetes [11].

Moreover, 5-HT2CR mutants exhibit increased physical activity and activity-related energy expenditure without changes in the basal metabolic rate [14]. An age-dependent reduction of the physical activity-related energy expenditure in addition to hyperphagia contributes to middle-age-onset of obesity in 5-HT2CR mutants [14]. Uncoupling protein-1 in brown adipose tissue-mediated thermogenesis does not contribute to the altered energy expenditure [10,15], whereas β3-adrenergic receptor expression in white adipose tissue, which is associated with decreased oxygen consumption, is lower in obese 5-HT2CR mutants [15]. Thus, chronic hyperphagia and decreased sympathetic neural activity in white adipose tissue in relation to the decreased activity-related energy cost may contribute to the middle age-onset of obesity in 5-HT2CR mutants.

### 2.2. The 5-HT2CR and 5-HT1BR Interaction in the CNS

5-HT2CR and 5-HT1BR are co-expressed in some hypothalamic nuclei and brainstem locations thought to be involved in ingestion and have their complementary effects on feeding circuits [2]. 5-HT1BRs locate on presynaptic neurons, and 5-HT2CRs locate on postsynaptic neurons [2].

Both 5-HT1BRs and 5-HT2CRs contribute to 5-HT-induced feeding suppression [2], but 5-HT1BR mutant mice do not exhibit the remarkable obese phenotype [16]. Systemic administration of meta-chlorophenylpiperazine (mCPP) induces feeding suppression in wild-type mice and the anorexic effect of mCPP was blunted in fasted 5-HT2CR-deficient mice [10]. A subsequent study, however, demonstrated that administration of mCPP suppressed food intake in fed 5-HT2CR-deficient mice as well as age-matched wild-type mice [17]. Moreover, the anorectic effects of mCPP and fenfluramine are attenuated in 5-HT1BR-deficient mice [18,19]. These findings suggest that 5-HT1BRs rather than 5-HT2CRs contribute to the anorectic effects induced by mCPP.

Systemic administration of a selective 5-HT1BR agonist leads to appetite suppression [20]. The hypophagia induced by the administration of a 5-HT1BR agonist is not attenuated by pharmacologic and/or genetic inactivation of 5-HT2CR [21], indicating that 5-HT2CRs do not likely mediate the anorexic effects of 5-HT1BR activation. Rather, 5-HT2CR mutant mice are actually more sensitive to the anorectic effects of the selective 5-HT1BR agonist CP 94253 than wild-type mice [21]. Pharmacologic inactivation of 5-HT2CRs leads to a rapid increase in hypothalamic 5-HT1BR gene expression [20]. Together, these findings suggest that 5-HT2CR downregulates the expression of hypothalamic 5-HT1BRs as well as the satiety signaling of 5-HT1BRs (Figure 1).

Selective serotonin re-uptake inhibitors (SSRIs), which were originally developed for the treatment of depression and panic disorders, are now also used in the treatment of binge-eating disorders and bulimia nervosa. The SSRI citapran induces increases in extracellular 5-HT levels in the hippocampus in rats, and while the selective 5-HT2CR antagonist SB 242084 enhances this effect, SB 242084 alone has no significant effects [22]. In addition, the effect of the SSRI fluoxetine to increase extracellular 5-HT levels in the prefrontal cortex is enhanced in 5-HT2CR-null mutant mice [22]. 5-HT2CR inactivation enhances the antidepressive effects of SSRIs [22]. The SSRI fluvoxamine has anorectic effects in mice via 5-HT1BRs in mice when 5-HT2CRs are pharmacologically inhibited, although fluvoxamine alone does not affect food intake [20]. Thus, the complemental interactions between 5-HT2C and 5-HT1BRs might contribute to regulate appetite.

## 3. The Central 5-HT and Neuropeptides Regulating Feeding Behavior

### 3.1. The 5-HT and POMC

5-HT2CRs are distributed on pro-opiomelanocortin (POMC) neurons in the arcuate nucleus (ARC) of the hypothalamus, suggesting the possibility that 5-HT2CRs and the melanocortin pathway interact in the regulation of feeding behavior [23,24]. Genetic deletion of 5-HT2CRs decreases the expression of hypothalamic POMC in association with hyperphagia [25], suggesting that POMC activity in the hypothalamus substantially contributes to the altered feeding behavior observed in 5-HT2CR mutants. Both d-fenfluramine and mCPP stimulate the activation of POMC neurons located in the ARC in vitro [23]. Consistent with this hypothesis, d-fenfluramine and mCPP activate α-melanocyte stimulating hormone (MSH)-containing neurons [23,24]. The putative transient receptor potential C (TRPC) channels are thought to mediate the activation of a subpopulation of POMC neurons by mCPP [26].

mCPP, however, can stimulate 5-HT1BRs and 5-HT2CRs. In regards to the network between 5-HT1BRs and POMC neurons, Heisler et al. suggested the following hypothesis for the hypothalamic mechanisms underlying the satiety-inducing effects of 5-HT1BR stimulation: 5-HT2CRs are expressed in POMC/cocaine- and amphetamine-related transcript (CART)-expressing neurons in the arcuate nucleus, whereas 5-HT1BRs are expressed in neuropeptide Y (NPY)/Agouti-related peptide (AgRP) neurons (24). Activation of the somatodendritic 5-HT1BRs leads to inhibition of NPY/AgRP neuronal activity, while the activation of presynaptic 5-HT1BRs expressed on the axon terminals of NPY/AgRP neurons suppresses the inhibition of POMC/CART neurons induced by γ-amino butyric acid [24].

On the other hand, Nonogaki et al. hypothesized that 5-HT1BRs directly upregulate POMC/CART neurons, leading to satiety [20]. Systemic administration of the selective 5-HT1BR agonist CP94253 leads to a rapid increase in the gene expression of hypothalamic POMC and CART, and a decrease in the gene expression of hypothalamic orexin, without altering NPY gene expression [20]. The altered expression of these hypothalamic neuropeptides is associated with the feeding suppression induced by administration of a 5-HT1BR agonist [20].

Melanocortin and β-endorphin neuropeptides are processed by POMC neurons. The melanocortins include adrenocorticotropic hormone (ACTH) and the different forms of MSH. α-MSH is the primary source of ligand for melanocortin-4 rceptor (MC4R) in the hypothalmus and has the inhibitory role in feeding and energy storage [13]. Genetic disruption of MC4R dispaly hyperphagia, hyperinsulinemia, and obesity [13]. Despite normal melanocortin signaling, β-endorphin-deficient mice display hyperphagia and obesity [27], suggesting that β-endorphin is required for normal feeding behavior. Both 5-HT2CR-deficient mice and 5-HT2CR-deficient mice with a heterozygous β-endorphin mutation (2CREnd mice) display hyperphagia [17]. 5-HT2CR-deficient mice exhibit decreased expression of hypothalamic POMC and NEFA/nucleobindin2 (NUCB2), whereas 5-HT2CR-deficient mice with a heterozygous β-endorphin mutation (2CREnd mice) exhibit decreased hypothalamic NUCB2 expression and increased hypothalamic POMC expression [25].

Systemic administration of mCPP suppresses food intake in 5-HT2CR mutants, 2CREnd mice, and age-matched wild-type mice [17]. On the other hand, the SSRI fluvoxamine has anorectic effects in 2CREnd mice [17], but does not affect food intake in age-matched wild-type mice or 5-HT2CR mutants [17]. These findings suggest that fluvoxamine-induced anorexia in mice requires perturbed 5-HT2CR and β-endorphin signaling plus functional hypothalamic POMC activity, whereas mCPP-induced anorexia does not always require functional 5-HT2CR and β-endorphin activity.

In addition to the ARC of hypothalamus, 5-HT2CRs are distributed on POMC neurons in nucleus of the solitary tract (NTS), but POMC expression is not abundant in the NTS [28]. Knockdown of functional POMC neurons in the NTS does not fully suppress the acute anorexic effects of the selective 5-HT2CR agonists such as lorcaserin and WAY161,503 within 2 h after the administration, whereas knockdown of 5-HT2CR in the ARC attenuated them [28]. These findings suggest that POMC neurons in the NTS less contribute to the 5-HT2CR-mediated feeding suppression than those in the hypothalamic ARC.

### 3.2. The 5-HT and MC4R

The MC4R is suggested as a crucial target of the 5-HT signaling of satiety. The initial report suggested that the anorectic action of fenfluramine is attenuated in A^y^ (yellow agouti) mice, which are deficient in melanocortin pathway signaling [23]. The appetite-suppressing effects of mCPP and fenfluramine were thought to be mediated by 5-HT2CR activation and the downstream melanocortin pathway [23].

In contrast to the initial findings, a study by Nonagaki et al. demonstrated that hyperphagic A^y^ mice are responsive to mCPP and fenfluramine-induced appetite suppression [29]. The anorectic effects of mCPP and fenfluramine are decreased, however, in food-restricted A^y^ mice, in which the expression of hypothalamic 5-HT1BRs and 5-HT2CRs is decreased [29]. Under restricted feeding conditions, the agouti peptide downregulates the gene expression of 5-HT2CRs and 5-HT1BRs in the hypothalamus, and the anorectic effects of mCPP and fenfluramine depend on the feeding condition in mice [29] (Table 2).

MC4R-deficient mice show an earlier onset of hyperphagia and obesity associated with hyperinsulinemia than 5-HT2CR-deficient mice [13,30]. The anorectic effects of fenfluramine or the selective 5-HT1BR agonist CP95253, however, are attenuated in lean and obese MC4R-deficient mice as well as in A^y^ mice [23,24], suggesting that MC4R may contribute to the feeding suppression induced by the activation of 5-HT1BRs.

In regards to interactions between 5-HT2CRs and MC4Rs in the regulation of feeding, a further report has showed that MC4R signaling in cholinergic neurons is not required for the anorexic effects of lorcaserin, a selective 5-HT2CR agonist, in mice, whereas MC4Rs in non-cholinergic neurons are required for the anorexic effect of lorcaserin [31]. In the study, a dose of locaserin (7.5 mg/kg) slightly suppressed food intake for 3 h in the wild-type mice, whereas the mild suppression of food intake induced by lorcaserin was attenuated in mice with genetic ablation of MC4Rs on non-cholinergic neurons [31]. The acute effects of lorcaserin within 2 h after the administration, the different doses of lorcaserin, and/or the chronic effects of lorcaserin on food intake and body weight in mice with genetic ablation of the MC4R, however, have not been evaluated.

On the other hand, MC4Rs in cholinergic neurons are required for the improvement of glucose homeostasis induced by lorcaserin [31]. The preclinical results suggested that lorcaserin has the glucose lowering effect independently of feeding suppression and weight loss. The central cholinergic MC4R pathways to peripheral organs involved in regulation of glucose metabolism remain to be elucidated. Further studies will be needed to confirm the results of the network between 5-HT and MC4Rs in the regulation of food intake and glucose metabolism.

### 3.3. The 5-HT and Orexin

Orexin neurons localize in the hypothalamus, including the lateral hypothalamus and perifornical area and the posterior hypothalamus [32,33,34]. Orexin is an orexigenic peptide that promotes wakefulness and decreases slow wave sleep and rapid eye movement sleep [32,33]. Orexin deficiency induces narcolepsy in humans and other mammalian species [34]. Orexin neurons receive inhibitory projections from the raphe nuclei. 5-HT neurons in the raphe nuclei are considered crucial for wakefulness and receive dense excitatory projections of orexin neurons.

A role for hypothalamic orexin in the regulation of 5-HT and POMC-mediated satiety is implicated. Intracerebroventricular injection of POMC small interfering-RNA oligonucleotides increases daily food consumption [35]. The effect is attenuated by intracerebroventricular injection of orexin small interfering-RNA oligonucleotides [35]. These findings suggest that POMC downregulates central orexin activity. The anorectic effects of mCPP require functional orexin activity and increased hypothalamic POMC activity [35] (Table 3).

On the other hand, 5-HT2CR mutants exhibit hyperphagia associated with decreased hypothalamic POMC expression [25] and increased hypothalamic orexin activity [2]. The compensatory increase in functional orexin activity may contribute to the increased wakefulness observed in 5-HT2CR mutants [2]. Thus, the balance of functional POMC and orexin neuronal activities may have a crucial role in the 5-HT2CR-mediated regulation of feeding.

### 3.4. The 5-HT and NUCB2

Nesfatin-1, processed by the cleavage of a precursor, nonessential fatty acids/nucleobindin2 (NUCB2), and distributed throughout the CNS, is implicated in the regulation of feeding by its actions on various nuclei, including the hypothalamic paraventricular nucleus, arcuate nucleus, lateral hypothalamus, and supraoptic nucleus [36,37]. Double-labeling immunohistochemistry in these areas revealed that nesfatin-1 colocalizes with POMC, CART, corticotropin-releasing hormone (CRH), and oxytocin [37,38]. NUCB2 is a novel satiety molecule associated with central melanocortin signaling, and intracerebroventricular injection of α-MSH, which is released from POMC neurons, increases the expression of hypothalamic NUCB2 [36]. In Zucker rats with a LepR mutation, food intake is decreased by intracerebroventricular injection of NUCB2, and leptin administration does not affect the expression of hypothalamic NUCB2 [36]. These findings suggest that the satiety signaling mechanism of NUCB2 is leptin-independent. The expression of hypothalamic NUCB2 induced by systemic administration of mCPP is significantly increased in wild-type mice, whereas it is attenuated in 5-HT2CR mutant mice [25].

Hyperphagic and non-obese 5-HT2CR mutants exhibit significantly decreased expression of hypothalamic NUCB2 and POMC in comparison with age-matched wild-type mice [25]. Despite the increased expression of hypothalamic POMC, however, 5-HT2CR mutant mice with heterozygous mutation of the β-endorphin gene have decreased hypothalamic NUCB2 expression [25]. Together, these findings suggest that 5-HT systems in the brain affect the expression of hypothalamic NUCB2 via 5-HT2CR activation. In the CNS, NUCB2 might be a specific downstream signaling target of 5-HT2CRs. As decreased expression of hypothalamic NUCB2 is not always associated with decreased POMC expression in 5-HT2CR mutant mice with a heterozygous mutation of the β-endorphin gene [25]. Thus, 5-HT2CR might contribute to the regulation of NUCB2 gene expression via a POMC-independent pathway.

NUCB2-induced feeding suppression is attenuated in MC4R-deficient mice, suggesting that MC4R is a target of NUCB2 as well as α-MSH [36]. Interestingly, central POMC activity is not essential for the 5-HT2CR-induced downregulation of NUCB2 expression. Together, these findings suggest that NUCB2 is a novel downstream pathway of 5-HT2CR with actions independent from those of POMC and α-MSH signaling (Table 3). Hypothalamic NUCB2 activity might be directly downregulated by 5-HT2CR activation, and this could contribute to the satiety signaling of MC4R. It is unclear, however, whether 5-HT2CRs are actually expressed on NUCB2 neurons in the hypothalamus and whether NUCB2 contributes to the anorexia induced by 5-HT2CR activation.

## 4. The Leptin and Brain 5-HT

### 4.1. Leptin and Brain 5-HT Synthesis

Leptin, an adipocyte-derived hormone, is involved in regulating food intake and body weight. Oury and Karsenty have suggested the biologic importance of leptin regulation of 5-HT [39]. Brain-derived 5-HT synthesis is initiated by the hydroxylation of tryptophan, a rate-limiting reaction performed by the enzyme TPH2 in the neurons of the dorsal and median raphe nuclei in the brainstem. The signaling form of the leptin receptor (LepR) is expressed in brainstem 5-HTergic neurons [40]. Peripheral injection of leptin into wild-type mice decreases brain TPH2 expression, whereas TPH2 expression and brainstem 5-HT content are increased in ob/ob mice with a leptin deficiency [40]. Neurophysiologic experiments using whole-cell patch recordings performed on brain slices revealed that leptin decreased the action potential frequency in 5-HTergic neurons of wild-type mice, but not in those of mice lacking LepR in neurons expressing TPH2 [40]. These findings suggest that leptin directly inhibits 5-HT production and release by the raphe nuclei neurons of the brainstem via LepR [39,40]. Moreover, removing one allele of TPH2 from ob/ob mice normalized the brain-derived 5-HT content, as well as the food intake and energy expenditure [40]. Despite the lack of leptin, removing both TPH2 alleles from ob/ob mice decreased food intake [40]. Based on these findings, Oury and Karsenty, and Yadav et al. suggested that leptin increases satiety and energy expenditure through the inhibition of 5-HT synthesis and release from brainstem neurons [39,40].

On the other hand, Lam et al. demonstrated that leptin does not influence appetite by direct actions on brain 5-HT neurons [41]. Although some LepR neurons lie close to 5-HT neurons in the dorsal raphe, 5-HT neurons do not express LepR. In addition, leptin hyperpolarizes some non-5HT dorsal raphe neurons, but does not alter the activity of dorsal raphe 5-HT neurons [41]. Moreover, pharmacologic depletion of 5-HT using p-chlorophenylalanine, a TPH inhibitor, does not interfere with the effect of leptin to attenuate food intake [41]. These findings suggest that brain 5-HT is not required for the anorectic actions of leptin. The discrepancy may be due to differences in the genetic and pharmacologic depletion of 5-HT. Genetic depletion of TPH2 might cause compensatory changes in central neurotransmitter systems other than 5-HT that influence leptin signaling. Thus, the leptin and brain 5-HT network remains to be elucidated.

### 4.2. The 5-HT2CR and LepR

5-HT2CRs and LepRs are both widely expressed in the hypothalamic regions that regulate energy intake and expenditure [2,11,13], but the developmental pattern of physiologic changes in 5-HT2CR and LepR mutants differs [11,13]. In 5-HT2CR mutant mice, the primary deficit is in the regulation of food consumption, and not glucose or fat metabolism [11]. Despite their normal body weight, 5-HT2C receptor mutants exhibit leptin-independent hyperphagia and hyperactivity at 2 to 3 months of age [11,14]. The changes in feeding behavior are not associated with changes in the plasma leptin, insulin, corticosterone, glucose, or lipid levels [11].

Chronic hyperphagia and hyperactivity lead to middle-age onset obesity and secondary insulin resistance in mice after 6 months of age. A high-fat diet enhances the development of obesity and insulin resistance, leading to type 2 diabetes in 5-HT2C receptor mutants [11]. Middle-age obesity in the 5-HT2C receptor mutants is associated with an age-dependent decrease in the energy cost of physical activity [11]. In addition, 2- to 3-month-old 5-HT2CR mutant mice exhibit significantly increased wakefulness and reduced slow wave sleep, but no effects on rapid eye movement sleep are observed [2]. These findings suggest a sleep loss-associated hyperphagia, which leads to late-onset obesity in animals with genetic ablation of 5-HT2C receptors.

Circulating leptin signals the CNS to rapidly increase sympathetic outflow and inhibit food intake [12,13]. LepR disturbances therefore disrupt both sympathetic outflow and satiety at the early stages. Hyperphagia, hyperinsulinemia, and decreased sympathetic outflow are observed in mice with disruption of the leptin gene (ob/ob) as well as in mice and rats with genetic disturbances of LepR signaling (db/db and fa/fa, respectively), and lead to early onset obesity [13]. In addition, ob/ob and db/db mice exhibit increased sleep time with increased short-wave sleep and decreased rapid eye movement sleep [42,43]. Expression of hypothalamic orexin and POMC is decreased in ob/ob and db/db mice [2]. The decreased orexin activity might contribute to the altered sleep-waking cycle in ob/ob and db/db mice. Thus, hyperphagia in 5-HT2C receptor mutants is associated with sleep loss, whereas hyperphagia in ob/ob and db/db mice is associated with increased sleep.

Systemic administration of mCPP suppresses food intake in LepR-deficient db/db mice [25], and systemic administration of leptin suppresses food intake in both young adult 5-HT2CR-deficient mice and wild-type mice [11]. Based on these findings, central LepR signaling and 5-HT1B/2CR signaling regulate feeding behavior and energy homeostasis via independent pathways.

## 5. The GLP-1 and 5-HT

### 5.1. GLP-1 and GLP-1R

Glucagon-like-peptide 1 (GLP-1) is mainly produced by the enteroendocrine L cells of the intestine and preproglucagon (PPG) neurons of the brain [44,45]. Gut-derived GLP-1 acts via GLP-1 receptors (GLP-1Rs) on peripheral organs including pancreatic islets and the gastrointestinal tract. Gut-derived GLP-1, however, is not required for the regulation of food intake and body weight [46]. Despite an almost complete reduction of circulating GLP-1 levels, food intake and body weight are normal in gut-derived GLP-1-deficient mice [47]. Brainstem-derived GLP-1, but not gut-derived GLP-1, contributes to regulate food intake and body weight [47].

GLP-1R agonists, which stimulate glucose-dependent insulin release and suppress glucagon secretion and gastric emptying, are used for the treatment of type 2 diabetes. GLP-1R agonists suppress food intake and reduce body weight. GLP-1R agonists such as liraglutide and semaglutide were recently approved as anti-obesity agents in addition to anti-diabetic agents in the US and other western countries.

GLP-1Rs are widely expressed in the CNS, including the hypothalamus and hindbrain, and contribute to the regulation of food intake and body weight [46]. Recent studies using liraglutide, a GLP-1R agonist, suggested that GLP-1Rs in the ARC of the hypothalamus contribute to feeding regulation in rats [48]. Injecting liraglutide into the third cerebral ventricle suppresses food intake and decreases body weight in a dose-dependent manner in mice [49], suggesting that hypothalamic nuclei around the third cerebral ventricle contribute to the suppressive effects of liraglutide on food intake and body weight.

Genetic ablation of GLP-1R has no effect on food intake or body weight gain in mice fed a chow diet [50]. Ablation of PPG neurons in the NTS, which produce GLP-1, also has no effect on food intake, energy expenditure, or body weight in mice fed chow diet [51,52], suggesting that the NTS PPG neurons are not essential for food intake or energy homeostasis. Moreover, NTS PPG neurons are not essential for the anorexic effects of GLP-1R agonists in mice [52]. PPG neurons in the NTS predominately receive vagal input from oxytocin-receptor-expressing vagal neurons rather than GLP-1R-expressing neurons [52].

### 5.2. The GLP-1 and POMC-MC4R

In rats, liraglutide stimulates POMC/CART neurons in the ARC of the hypothalamus in vitro [48], and central POMC/CART-expressing ARC neurons are suggested to mediate the suppressive effects of liraglutide on food intake and body weight [48]. In mice, however, hypothalamic POMC/CART and MC4R pathways are not required for the acute suppressive effects of liraglutide on food intake and body weight [53,54]. Systemic administration of liraglutide does not significantly affect hypothalamic POMC or CART expression, whereas it acutely suppresses hypothalamic orexin expression in C57BL6J mice [54]. In addition, liraglutide suppresses food intake and body weight in heterozygous MC4R mutants [53], db/db mice with leptin R mutation, which have decreased hypothalamic POMC and CART expression [54], and freely fed KK mice, which show inherently glucose intolerance and insulin resistance [54], whereas the inhibitory effects of liraglutide on food intake and body weight are attenuated in food-restricted KK mice [54], leading to increased hypothalamic orexin expression [35]. Moreover, in obese humans with MC4R mutations, liraglutide causes weight loss [55,56]. These findings suggest that MC4Rs are not required for liraglutide-induced feeding suppression and weight loss in mice or humans.

### 5.3. The GLP-1 and 5-HT in the CNS

How interactions between central GLP-1 and 5-HT signaling affect food intake and body weight remains unclear. A report by Anderberg et al. suggested that brain-derived 5-HT is critical for maintaining weight loss induced by GLP-1 receptor activation in rats [57]. Pharmacologic inhibition of 5-HT synthesis, however, does not affect the acute anorexic effects of liraglutide in mice [54], and does not affect the feeding suppression induced by liraglutide for 24 h after injection in mice [58]. Moreover, pharmacologic inhibition of 5-HT synthesis does not affect acute weight loss or maintenance of weight loss induced by liraglutide in mice [54,58]. Thus, the feeding suppression and weight loss induced by liraglutide could be independent of 5-HT synthesis in mice. In addition, peripheral injection of liraglutide acutely suppresses food intake and body weight in 5-HT2CR mutant mice as well as in wild-type mice [53], suggesting that 5-HT2CRs are not required for the liraglutide-induced feeding suppression and weight loss in mice.

Anderberg et al. also suggested that pharmacologic blockade of central 5-HT2A receptors using R 96544 attenuates the chronic anorexic and weight loss effects of central injection of exedin-4 or intraperitoneal injection of liraglutide in rats [57]. On the other hand, liraglutide downregulates the expression of hypothalamic 5-HT2A receptors in fed mice [58]. Pretreatment with the selective 5-HT2AR agonist partially reverses the acute anorexic effects of liraglutide in mice [58]. Thus, decreased hypothalamic 5-HT2AR signaling may partially contribute to the acute anorexic effects of liraglutide in mice. The differences between central GLP1R signaling in POMC-MC4R pathways and 5-HT systems in the regulation of food intake and body weight may be species-specific.

On the other hand, 5-HT2CRs and 5-HT3Rs are expressed on PPG neurons [59] and PPG neurons receive 5-HTergic input and respond directly to 5-HT via a 5-HT2CR-dependent increase in dendritic [Ca^2+^] [60]. 5-HT acts via 5-HT2CR to stimulate the NTS PPG neurons, which may contribute to the inhibitory effects of 5-HT on food intake and body weight. The network between 5-HT and PPG neurons in the NTS requires further clarification.

## 6. Peripheral 5-HT Network in Feeding Signals

### 6.1. Nutrient Metabolites and Gut-Derived 5-HT

In the periphery, circulating 5-HT is primarily synthesized from tryptophan by Tph1 within EC cells of the gastrointestinal tract [61,62,63,64,65,66]. The expression and activity of Tph1 in EC cells are regulated by the actions of surrounding cells and nutrients [63,66]. In mice, plasma 5-HT levels are increased by fasting; feeding on a high carbohydrate diet that includes glucose, fructose, and sucrose; and feeding on a high fat diet [61,62,63,64,65,66,67,68,69]. A report suggested that nutrient metabolites stimulate 5-HT secretion from EC cells indirectly through GLP-1 released from enteroendocrine (EE) cells in the small intestine [63]. G-protein-coupled receptors (GPCRs) in the EE cells sense different nutrient metabolites [63]. The GPCRs in EE cells include free fatty acid receptor 1 (FFAR1)/G-protein-coupled receptor 40 (GPR40) sensing long-chain fatty acids, GPR119 sensing 2-acylglycerol, G-protein-coupled bile acid receptor 1 (GPBAR1)/TGR5 sensing bile acids, and calcium-sensing receptor (CasR) and GPR142 sensing amino acids and oligopeptides [63]. These nutrient-sensing GPCRs are absent in EC cells from the small intestine, whereas GLP-1R are highly expressed on EC cells [63]. Thus, the EE cells stimulate GLP-1 secretion in response to nutrient metabolites. The released GLP-1 binds the GLP-1Rs on EC cells and stimulates 5-HT secretion [63].

### 6.2. Microbial Metabolites and Gut-Derived 5-HT

In the colon, gut microbes regulate 5-HT synthesis and circulating 5-HT levels [63,64,65,66]. Spore-forming bacteria from mouse and human microbiota promote 5-HT biosynthesis from colonic EC cells [63,66]. In the colon, EC cells express different types of receptors for microbial metabolites including FFAR2/GPR132 sensing lactate and acyl amides, olfactory receptor 78 (OLF78) and OLF558 sensing different types of short chain fatty acids, GPBAR1/TGR5 sensing secondary bile acids, GPR35 sensing small aromatic acids and GPR132 sensing lactate and acyl amides [63]. Colonic EC cells express GLP-1Rs and other gut hormone receptors, such as those for glucose-dependent insulinotropic polypeptide and peptide YY [63].

### 6.3. The GLP-1 and 5-HT in the Gut

Administration of a selective 5-HT1BR agonist or selective 5-HT4R agonist increases plasma active GLP-1 levels and GLP-1-mediated insulin secretion, leading to improved glucose tolerance in mice [70,71]. Dipeptidyl peptidase-4 rapidly degrades the active form of GLP-1 (amino acids 7–36) to an inactive, N-terminal truncated form (amino acids 9–36) in the bloodstream. Pharmacologic stimulation of 5-HT1BRs or 5-HT4Rs increases plasma active GLP-1 and insulin levels and improves glucose tolerance under pharmacologic inhibition of dipeptidyl peptidase-4 in mice [70,71]. These findings suggest that gut-derived GLP-1 upregulates 5-HT synthesis and release in EC cells, and that 5-HT stimulates GLP-1 secretion from EE cells via 5-HT1BR or 5-HT4R in the small intestine. Thus, gut-derived GLP-1 in EE cells and 5-HT in EC cells can interact and contribute to glucose homeostasis (Table 2).

### 6.4. Metabolic Diseases and Gut-Derived 5-HT

In addition to feeding on high fat or high carbohydrate diets, plasma 5-HT levels are associated with obesity, non-alcoholic fatty liver diseases (NAFLD), type 2 diabetes [67,69,72,73,74], and cardiovascular disease (CVD) [75] in rodents and humans. Genetic, pharmacologic and nutrimental suppression of circulating 5-HT levels via Tph1 can protect against high-fat diet-induced metabolic diseases in mice [67,69,74]. Moreover, antibiotic-induced changes in the microbiota composition improves glucose tolerance via inhibiting synthesis of 5-HT in the gut [76]. Thus, inhibition of gut-derived 5-HT synthesis and microbiota depletion may have similar effects on glucose metabolism.

These findings suggest that increased peripheral 5-HT synthesis induced by nutrients such as by overeating, a high-fat diet, and a high carbohydrate diet, or altered microbiota composition may contribute to the pathophysiologic mechanisms of obesity-related diseases including metabolic syndrome, NAFLD, type 2 diabetes, and CVD, and the suppression of increased peripheral 5-HT synthesis may be a novel therapeutic approach for diseases related to obesity.

## 7. Peripheral 5-HT and FGF21

### 7.1. Metabolic Diseases and FGF21

Fibroblast growth factor 21 (FGF21) is widely expressed in various organs, including the liver, pancreas, skeletal muscle, and adipose tissues, but circulating FGF21 in diet-induced obesity and type 2 diabetes is liver-derived [77]. Although FGF21 has several beneficial effects for glucose and lipid metabolism [78], circulating FGF21 levels are paradoxically increased in humans with obesity, metabolic syndrome [79,80], and NAFLD [81,82,83]. Plasma FGF21 levels correlate with the severity of non-alcoholic steatohepatitis in subjects with obesity and type 2 diabetes [83].

In addition, plasma FGF21 levels are increased in subjects with prediabetes and type 2 diabetes [84,85,86], and correlate with hepatic and muscle insulin resistance in type 2 diabetes [84]. Plasma FGF21 levels could be an independent predictor of metabolic syndrome and type 2 diabetes in Caucasian [87] and Chinese subjects [88]. Moreover, higher serum FGF21 levels are associated with an increased risk of CVD [89,90,91].

### 7.2. Nutrients and the FGF21

Plasma FGF21 levels can be altered in response to nutrient intake. Fasting as well as intake of a high carbohydrate diet, including glucose and fructose, high fat, and low proteins increase plasma FGF21 levels in rodents [69,92,93,94,95,96,97]. A recent report demonstrated that the increases in plasma FGF21 levels and hepatic FGF21 expression precede hyperinsulinemia, insulin resistance, impaired glucose tolerance, and weight gain in mice fed a high-fat diet [69]. Ingestion of whey protein isolate suppresses hepatic FGF21 expression and plasma FGF21 levels in mice fed either a high fat diet or chow diet [69], whereas intake of β-conglycinin, a soy protein, increases hepatic FGF21 expression and plasma FGF21 levels in mice fed a high-fat diet [98].

### 7.3. The 5-HT and Liver-Derived FGF21

A recent study suggested that peripheral 5-HT upregulates hepatic FGF21 expression and circulating FGF21 levels [69]. Nutritional or genetic inhibition of circulating 5-HT levels suppresses plasma FGF21 levels and hepatic FGF21 expression in mice [69] (Figure 1). Activating transcriptional factor 4 may be involved as the transcriptional factor of peripheral 5-HT-induced hepatic FGF21 expression in mice [69]. Moreover, 5-HT2ARs and 5-HT2BRs may contribute to the 5-HT-mediated hepatic FGF21 expression and circulating FGF21 levels in mice [69,99].

5-HT2BRs are highly expressed in the liver and their expression increases upon fasting [68]. Gut-derived 5-HT promotes hepatic gluconeogenesis and inhibits glucose uptake through 5-HT2BRs in hepatocytes [68]. Pharmacologic inhibition of 5-HT2BRs ameliorates hyperglycemia associated with alterations of hepatic FGF21 and 5-HT2AR expression and plasma FGF21 levels without affecting food intake and body weight in obese and diabetic mice such as KKA^y^ and db/db mice [99].

On the other hand, recent studies demonstrated that hepatic expression of 5-HT2ARs is increased in obese mice fed a high-fat diet for 13 days [69] and 8 weeks [67], and either genetic ablation of liver-specific 5-HT2ARs or treatment with a selective 5-HT2AR antagonist suppresses hepatic steatosis in mice fed a high-fat diet [67]. 5-HT2ARs in the liver may therefore contribute to hepatic steatosis in mice fed a high-fat diet [67]. Moreover, treatment with a selective 5-HT2AR agonist increases the expression of hepatic FGF21 and Sdf2l1, which are indicators of endoplasmic reticulum stress in mice [69], suggesting that 5-HT2ARs upregulate hepatic FGF21 and Sdf2l1 expression. Thus, 5-HT via 5- 5-HT2ARs and 5-HT2BRs may contribute to the regulation of hepatic FGF21 expression and plasma FGF21 levels.

**Figure 1 ijms-23-01600-f001:**
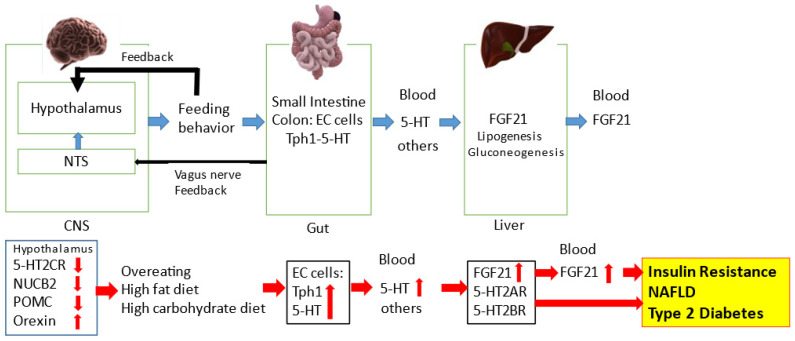
The regulatory role of the central and peripheral 5-HT network on feeding signals in metabolic diseases. The NTS and hypothalamus can regulate feeding behavior. Feeding behavior can alter the gut-derived 5-HT synthesis via Tph1 and circulating 5-HT levels. Circulating 5-HT and other factors can modulate hepatic FGF21 expression and circulating FGF21 levels. Feeding condition can alter hypothalamic neurotransmission including 5-HT2CR and 5-HT1BR signaling. Microbial metabolites and the gut-derived peptides can modulate central neurotransmission via the afferent vagus nerve to the NTS. Disruption of central 5-HT2CR signaling can decrease hypothalamic POMC and NUCB2 activity and increase hypothalamic orexin activity, leading to overeating. Increased energy intake via a high-fat diet and/or high-carbohydrate diet can upregulate the gut-derived 5-HT synthesis and circulating 5-HT levels. Circulating 5-HT can upregulate hepatic FGF21 expression and circulating FGF21 levels, which can precede hyperinsulinemia, insulin resistance, type 2 diabetes and NAFLD. Moreover, hepatic 5-HT2AR and 5-HT2BR signaling may be involved in hepatic FGF21 production and the pathophysiological mechanisms of NAFLD and type 2 diabetes.

## 8. Conclusions

Central 5-HT network systems via 5-HT2CRs regulate feeding behavior. Increased energy intake via a high-fat diet and/or high-carbohydrate diet can upregulate gut-derived 5-HT synthesis via Tph1. Gut-derived 5-HT can upregulate hepatic FGF21 expression and plasma FGF21 levels, which precede hyperinsulinemia, insulin resistance, impaired glucose tolerance and weight gain in mice fed a high-fat diet. In humans, increased levels of plasma FGF21 can predict metabolic syndrome and type 2 diabetes in healthy subjects, and are associated with insulin resistance, metabolic syndrome, type 2 diabetes, NAFLD and an increased risk of CVD. Thus, the effects of the central and peripheral 5-HT networks on feeding signals may contribute to the pathophysiologic mechanisms of obesity-related metabolic diseases. Modulation of the 5-HT network may be a novel preventive or therapeutic approach for treating obesity and obesity-related diseases.

## Figures and Tables

**Table 1 ijms-23-01600-t001:** 5-HTRs, structure and signal transduction.

5-HTRs	Structure	Signal Transduction	
5-HT2AR, 5-HT2BR, 5-HT2CR	GPCR	phospholipase C	IP3, DAG, PKC
5-HT1AR, 5-HT1BR	GPCR	AC ↓ cAMP ↑	PKA
5-HT4R, 5-HT6R, 5-HT7R	GPCR	AC ↓ cAMP ↑	PKA

GPCR, G-protein coupled receptor; cAMP, cyclic adenosine monophosphate; AC, adenylate cyclase; IP3, inositol triphosphate; DAG, diacylglycerol; PKC, proteinkinase C; PKA, proein kinase A.

**Table 2 ijms-23-01600-t002:** Feeding condition can alter hypothalamic 5-HT2CR and 5-HT1BR expression and responses to mCPP or fenfluramine administration in A^y^ mice.

Feeding Condition	Freely Feeding	Freely Feeding	Restricted Feeding
Age	5-wk-old	8-wk-old	5-wk-old
Food intake	Increase	Increase	Normal
Body weight	Normal	Increase	Decrease
Hypothalamic 5-HT2CR mRNA	Increase	Increase	Decrease
Hypothalamic 5-HT1BR mRNA	Increase	Increase	Decrease
Blood glucose	Increase	Increase	Decrease
Effect of mCPP on food intake	Suppression	Suppression	No effect
Effect of fenfluramine on food intake	Suppression	Suppression	No effect

**Table 3 ijms-23-01600-t003:** Interactions between central 5-HT2CR and neuropeptides regulating feeding signals.

5-HTRs	Neuropeptides	Effect of 5-HTRs	Site	Effects of Neuropeptides on Food Intake
5-HT2CR 5-HT1BR	POMC	Upregulation	ARC, NTS	Anorexic effect
5-HT2CR 5-HT1BR	Orexin	Downregulation	LHA	Orexigenic effect
5-HT2CR	NUBC2	Upregulation	ARC, LHA, PVH	Anorexic effect

ARC, arcuate nucleus; NTS, nucleus of the solitary tract; LHA, lateral hypothalamuic area; PVH, paraventricular nucleus of the hypothalamus.

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
