# Peer review of "The Regulatory Role of the Central and Peripheral Serotonin Network on Feeding Signals in Metabolic Diseases"

_ijms, 2022, doi:10.3390/ijms23031600_

Round 1
Reviewer 1 Report
Serotonin (5-HT) is involved in regulation of feeding and general metabolism. PubMed shows about 2900 publications in response to keywords 5-HT and feeding. At the same time, besides 5-HT there are numerous molecular factors regulating feeding and general metabolism such as melanocortins, orexins, leptin, GLP-1 etc. In this review the author tries to elucidate the place of 5-HT in the central and peripheral molecular networks regulating feeding. This idea is interesting and it has a great fundamental and medical impacts.
However, the huge amounts of publications concerning the regulation of feeding and general metabolism as well as the great molecular diversity of this regulation made this task very hard for one specialist. Moreover, the seven small topics in this review can be expanded to independent reviews.
As a result, the author discussed mainly the involvement of 5-HT2C in feeding and general metabolism regulation, while other 5-HT-related molecules, such as TPH2, TPH1, 5-HT1B, 5-HT2A, 5-HT2B receptors etc. are discussed seldom or not at all. At the same time, in PubMed there are about 400 publications concerning TPH2, TPH1, SERT and 5-HT receptors in feeding regulation. PubMed shows about 2000 publications in response to keywords 5-HT and melanocortins, 5-HT and orexins, 5-HT and leptin etc., while the author discusses only a small part of these publications corresponding mainly to 5-HT2C receptors. It seems that the real idea of this review is the role that 5-HT2C receptors play in the brain – gut - liver network regulating feeding.
Major concerns
- The review title is too general and, therefore, it should be corrected. I think that a title “The role that 5-HT2C receptors play in the brain-gut-liver network regulating feeding signals” (or similar) is more relevant. If the author will not decide to change the title (and the main idea), he has to strongly justify why he chose these particular publications for his review and ignored most of the available publications.
- The molecular biology, distribution and functions of 5-HT2C receptors are poorly described. It makes review difficult to read and understand for the physiologists who are not specialists in 5-HT system.
- 5-HT1B receptors on presynaptic 5-HT neurons cannot stimulate 5-HT release (line 94).
- There is no description of transgenic mice (lines 71, 294).
- The description of the key molecules involved in the network is not enough for scientists who are not specialists in melanocortins, leptin, etc.
Minor concerns
- The author has to decipher some acronyms, KK, GPR40, GPR35, OLF78 etc.
- What do the phrases “5-HT and POMC network”, “5-HT and MC4R network” etc mean?. May be they mean “5-HT and POMC interaction” etc.
- What does “non-cholinergic MC4Rs” (line 202) mean?
Author Response
Respond to Reviewer 1
Serotonin (5-HT) is involved in regulation of feeding and general metabolism. PubMed shows about 2900 publications in response to keywords 5-HT and feeding. At the same time, besides 5-HT there are numerous molecular factors regulating feeding and general metabolism such as melanocortins, orexins, leptin, GLP-1 etc. In this review the author tries to elucidate the place of 5-HT in the central and peripheral molecular networks regulating feeding. This idea is interesting and it has a great fundamental and medical impacts.
However, the huge amounts of publications concerning the regulation of feeding and general metabolism as well as the great molecular diversity of this regulation made this task very hard for one specialist. Moreover, the seven small topics in this review can be expanded to independent reviews.
As a result, the author discussed mainly the involvement of 5-HT2C in feeding and general metabolism regulation, while other 5-HT-related molecules, such as TPH2, TPH1, 5-HT1B, 5-HT2A, 5-HT2B receptors etc. are discussed seldom or not at all. At the same time, in PubMed there are about 400 publications concerning TPH2, TPH1, SERT and 5-HT receptors in feeding regulation. PubMed shows about 2000 publications in response to keywords 5-HT and melanocortins, 5-HT and orexins, 5-HT and leptin etc., while the author discusses only a small part of these publications corresponding mainly to 5-HT2C receptors. It seems that the real idea of this review is the role that 5-HT2C receptors play in the brain – gut - liver network regulating feeding.
Ans. We feel that the reviewer 1 cannot understand the essence of this review paper exactly in this special issue “ Serotonin network and energy metabolism”. This is not a book for general readers and medical students, but an overview of the special issue. All information of 5-HT and feeding signals do not need to include. We feel that the essential papers including the original articles and review papers are well included. The major researchers are very limited in this field. This review paper is based on my own original articles and review papers over 25 years of this research field, and major publications by my colleagues and major researchers in this field. After reading this review, some researchers may have a new idea of research and may submit the original or independent review papers to expand each small topics. That will be very helpful and constructive in this special issue of IJMS.
Because the excellent review papers of “5-HT2CR and energy homeostasis” or “peripheral 5-HT and energy metabolism” are reported, we avoided overlapping the basic information as much as possible and provided the essential background of 5-HT2CR involved in regulating feeding and energy metabolism, and provided a recent novel concept of the gut-derived 5-HT-mediated hepatic FGF21 production, which may lead to obesity and obesity-related diseases. This is a critical point of this review paper and there are no review reports elsewhere. Thus, this review is not limited in 5-HT2CRs, which have no reports to show the direct effects on the gut and liver.
Response to Major concerns
- The review title is too general and, therefore, it should be corrected. I think that a title “The role that 5-HT2C receptors play in the brain-gut-liver network regulating feeding signals” (or similar) is more relevant. If the author will not decide to change the title (and the main idea), he has to strongly justify why he chose these particular publications for his review and ignored most of the available publications.
Ans. We think that the comments by the reviewer 1 that “The review title is too general” are not appropriate. 5-HT2CRs are highly expressed in the CNS, but there are no reports to show the direct effects of 5-HT2CRs on the gut and liver. From the basic background, we disagree with the reviewer 1’s suggestion. We changed the title to “The regulatory role of the central and peripheral serotonin network on feeding signals in metabolic diseases” in the revised paper based on the reviewer 1, 2 and 3’s comments.
- The molecular biology, distribution and functions of 5-HT2C receptors are poorly described. It makes review difficult to read and understand for the physiologists who are not specialists in 5-HT system.
Ans. The basic information of 5-HT2CRs are well known in many reports by other investigators and my own previous reviews. In the revised paper, we added the brief information of the molecular biology, distribution and functions of 5-HTRs regulating feeding signals in the Table 1, 2 and 3.
- 5-HT1B receptors on presynaptic 5-HT neurons cannot stimulate 5-HT release (line 94).
Ans. 5-HT1BRs locate on presynaptic neurons (where they stimulate 5-HT release). we excluded (where they stimulate 5-HT release) in the revised paper.
- There is no description of transgenic mice (lines 71, 294).
Ans, There are no reports of 5-HT2CR transgenic mice, which show altered feeding behavior and energy metabolism.
- The description of the key molecules involved in the network is not enough for scientists who are not specialists in melanocortins, leptin, etc.
Ans. Leptin is well described in the original text, but we added a brief explanation of leptin in the Introduction and melanocortins according to the reviewer 1 suggestion in the revised paper.
Minor concerns
- The author has to decipher some acronyms, KK, GPR40, GPR35, OLF78 etc.
Ans. We added decipher some acronyms as the reviewer 1 suggested. Because KK is a mouse strain name, we cannot decipher it. We added the brief explanation of KK strain, “which shows inherently glucose intolerance and insulin resistance” in the revised paper.
- What do the phrases “5-HT and POMC network”, “5-HT and MC4R network” etc mean?. May be they mean “5-HT and POMC interaction” etc.
Ans. We excluded “network” in the phrases as the reviewer suggested.
- What does “non-cholinergic MC4Rs” (line 202) mean?
Ans. It means “MC4Rs in non-cholinergic neurons”. It was changed in the revised paper.
Reviewer 2 Report
Title: Serotonin network in the brain-gut-liver axis regulates feeding 2
signals
Author keenly presented the Central and peripheral 5-HT signaling and their key impact on obesity and type-2 diabetes.
He utilized the crucial network concepts such as NUCB2 network, orexin network, MC4R network, POMC network, Leptin, and GLP-1 networks with 5-HT clarified the role in various diseases (metabolic syndrome, type 2 diabetes, NAFLD and an increased risk of CVD etc).
This review more useful for a novel preventive or therapeutic approach for treating obesity and obesity-related diseases through modulate the 5-HT network.
Author Response
Response to Reviewer2
Author keenly presented the Central and peripheral 5-HT signaling and their key impact on obesity and type-2 diabetes.
He utilized the crucial network concepts such as NUCB2 network, orexin network, MC4R network, POMC network, Leptin, and GLP-1 networks with 5-HT clarified the role in various diseases (metabolic syndrome, type 2 diabetes, NAFLD and an increased risk of CVD etc).
This review more useful for a novel preventive or therapeutic approach for treating obesity and obesity-related diseases through modulate the 5-HT network.
Ans. Thank very much you for your useful and constructive comments.
Reviewer 3 Report
I have reviewed the manuscript "Serotonin network in the brain-gut-liver axis regulates feeding signals" submitted by Katsunori Nonogaki to IJMS (ijms-542020).
I must agree that the topic is worthy of scientific attention and may have multiple implications in clinical praxis. It is a review article and the author does not present any new data - which is OK for this purpose. The author has obviously an expert insight into this issue. However, I also believe that it should be possible to prepare for a review illustrations and schemes of significantly higher quality. As it is, I do not really believe that this is eye-catching to anyone.
The body of the manuscript is very conventionally divided and follows the background of the 5HT role in organisms. There is nothing wrong with that. The review of central, peripheral and enteric 5HT roles seems to be prepared relatively well.
However, the author primarily aimed to summarise recently available scientific knowledge on "brain-gut-liver axis" (I see it in the headline). If so, I miss here a critically important section carefully describing mechanistically also the physical aspects of this axis. 5HT is, indeed, present in different tissues and exerts different roles. However, we urgently need to see interconnections of these structures. The author completely omited the feedback signals (!), the potential role of the blood-brain barrier (as it acts as e.g. an obstacle for various pharmacologicall interventions). I believe that anatomical insight into this "axis" would be very helpful to readers. I also believe that this can help to separate correctly structural evidence from functional speculations.
Further, FGF21 is an immensely important topic. It is also known as a myokine and there are other molecules from this group (namely IL-6) that can regulate FGF21 and also modify e.g. food intake (e.g. in depression, in cancer cachexia etc.). I believe that the presented FGF21 facts should be presented more carefully and broader view.
To conclude, the presented manuscript is promising - however, it is not finished yet. I believe that there is still some time necessary for perfecting this work and resubmitting the review in a more satisfying form.
Author Response
Response to Reviewer 3
I have reviewed the manuscript "Serotonin network in the brain-gut-liver axis regulates feeding signals" submitted by Katsunori Nonogaki to IJMS (ijms-542020).
I must agree that the topic is worthy of scientific attention and may have multiple implications in clinical praxis. It is a review article and the author does not present any new data - which is OK for this purpose. The author has obviously an expert insight into this issue. However, I also believe that it should be possible to prepare for a review illustrations and schemes of significantly higher quality. As it is, I do not really believe that this is eye-catching to anyone.
Ans. We revised the Figures according to the reviewer 3’s suggestion, and changed the original Figures to Table 1-3 and a new Figure 1 in the revised paper.
The body of the manuscript is very conventionally divided and follows the background of the 5HT role in organisms. There is nothing wrong with that. The review of central, peripheral and enteric 5HT roles seems to be prepared relatively well.
Ans. We appreciate the comments by the reviewer 3, but the above comments and the score 1 regarding with the first editorial question are not consistent.
However, the author primarily aimed to summarise recently available scientific knowledge on "brain-gut-liver axis" (I see it in the headline). If so, I miss here a critically important section carefully describing mechanistically also the physical aspects of this axis. 5HT is, indeed, present in different tissues and exerts different roles. However, we urgently need to see interconnections of these structures. The author completely omited the feedback signals (!), the potential role of the blood-brain barrier (as it acts as e.g. an obstacle for various pharmacologicall interventions). I believe that anatomical insight into this "axis" would be very helpful to readers. I also believe that this can help to separate correctly structural evidence from functional speculations.
Ans. We excluded the word “axis” and changed the Title in the revised paper, but the word “axis” does not always need to include “the feedback signals” as the reviewer 3 suggested (see Ref. 67 in the original text: Title: Serotonin signals through a gut-liver axis to regulate hepatic steatosis. Nat Commun 2018.).
Moreover, circulating FGF21 within physiological ranges does not affect food intake and body weight. This is similar to circulating GLP-1 in vivo.
We added Table 2, which show altered feeding condition can modulate central 5-HT2C/1BR expression and responses to the agonists, and feedback by feeding behavior and the afferent vagus nerve from the gut in Figure 1 in the revised paper.
Further, FGF21 is an immensely important topic. It is also known as a myokine and there are other molecules from this group (namely IL-6) that can regulate FGF21 and also modify e.g. food intake (e.g. in depression, in cancer cachexia etc.). I believe that the presented FGF21 facts should be presented more carefully and broader view.
Ans. Excellent review papers about FGF21 are reported by other researchers. Circulating FGF21 is mainly derived from the liver but not muscle. The aim of this review paper is to provide an overview of my own special issue “5-HT network and energy metabolism” and to present the regulatory role of the central and peripheral 5-HT network on feeding signals in obesity and obesity-related diseases. Cytokines, the mediators of infection and inflammation, depression and cancer cachexia are different subjects. Because the aims and focus of this review paper are unclear, I will not treat the broader topics as the overview in this special issue. The central 5-HT2CR signaling and the gut-derived 5-HT-liver derived FGF21 interaction as feeding signals in metabolic diseases are the main text in this review paper.
To conclude, the presented manuscript is promising - however, it is not finished yet. I believe that there is still some time necessary for perfecting this work and resubmitting the review in a more satisfying form.
Round 2
Reviewer 3 Report
Sir,
I have reviewed the updated version of the manuscript ijms-1542020, Serotonin network in the brain-gut-liver axis regulates feeding signals submitted by Katsunori Nonogaki.
The author provided subtle text changes including the headline. I believe the headline sounds more realistic now (The regulatory role of the central and peripheral serotonin network on feeding signals in metabolic disease). At least, the headline is not misleading anymore.
We can philosophically dispute the necessity of feedback control mechanisms in biomedical science. However, I believe that it is scientifically sound to highlight these regulatory aspects. But it exceeds the scope of the submitted manuscript. Due to the headline change, it is satisfactory.
As far as I see, Figure 3 gained from the first version some complexity. However, the Legend in this figure also requires some enhancement to increase clarity for novice readers.
Figures 1 and 2 were converted into simplistic tables - but it was also somewhat helpful.
Line 429 - The headline starts with a dot .... please, format this.
To conclude, the submitted text represents a review of a potentially clinically relevant issue and it can be useful for interested readers.
Author Response
Response to Reviewer 3
I have reviewed the updated version of the manuscript ijms-1542020, Serotonin network in the brain-gut-liver axis regulates feeding signals submitted by Katsunori Nonogaki.
The author provided subtle text changes including the headline. I believe the headline sounds more realistic now (The regulatory role of the central and peripheral serotonin network on feeding signals in metabolic disease). At least, the headline is not misleading anymore.
We can philosophically dispute the necessity of feedback control mechanisms in biomedical science. However, I believe that it is scientifically sound to highlight these regulatory aspects. But it exceeds the scope of the submitted manuscript. Due to the headline change, it is satisfactory.
As far as I see, Figure 3 gained from the first version some complexity. However, the Legend in this figure also requires some enhancement to increase clarity for novice readers.
Ans. We added the Legend in Figure 1 according to the reviewer 3 suggestion in the revised paper. Because this Figure is a summary of the main text and the conclusion also includes the same content, this legend is overlapped.
Figures 1 and 2 were converted into simplistic tables - but it was also somewhat helpful.
Line 429 - The headline starts with a dot .... please, format this.
Ans. We could not find the point as the reviewer suggested in line 429 of the revised paper. The dot might be added by mistake during the format conversion. The lay out team will format this manuscript further after acceptance.
To conclude, the submitted text represents a review of a potentially clinically relevant issue and it can be useful for interested readers.